# RETRACTED: Protein Tyrosine Phosphatase Non-Receptor 11 (*PTPN11*/Shp2) as a Driver Oncogene and a Novel Therapeutic Target in Non-Small Cell Lung Cancer (NSCLC)

**DOI:** 10.3390/ijms241310545

**Published:** 2023-06-23

**Authors:** Cathy E. Richards, Yasir Y. Elamin, Aoife Carr, Kathy Gately, Shereen Rafee, Mattia Cremona, Emer Hanrahan, Robert Smyth, Daniel Ryan, Ross K. Morgan, Susan Kennedy, Lance Hudson, Joanna Fay, Kenneth O’Byrne, Bryan T. Hennessy, Sinead Toomey

**Affiliations:** 1Medical Oncology Group, Department of Molecular Medicine, Royal College of Surgeons in Ireland, D09 YD60 Dublin, Ireland; bryanhennessy74@gmail.com (B.T.H.); sineadtoomey@rcsi.ie (S.T.); 2Department of Thoracic Head and Neck Medical Oncology, Division of Cancer Medicine, M.D. Anderson Cancer Centre, Houston, TX 77030, USA; 3Thoracic Oncology Research Group, Trinity Translational Medicine Institute, Trinity College Dublin, St. James’s Hospital, D08 NHY1 Dublin, Ireland; 4Department of Medical Oncology, St. Vincent’s Hospital, D04 T6F4 Dublin, Ireland; 5Department of Respiratory Medicine, Beaumont Hospital, D09 V2N0 Dublin, Ireland; rossmorgan@beaumont.ie; 6Department of Pathology, St. Vincent’s Hospital, D04 T6F4 Dublin, Ireland; 7Department of Surgery, Royal College of Surgeons in Ireland, D09 YD60 Dublin, Ireland; lhudson@rcsi.ie; 8RCSI Biobank Service, Royal College of Surgeons in Ireland, D09 YD60 Dublin, Ireland; 9Princess Alexandra Hospital, Brisbane, QLD 4102, Australia

**Keywords:** Shp2, *PTPN11*, lung cancer, somatic mutations, PI3K signalling pathway, MAPK signalling pathway, cancer therapy, targeted therapy, anti-cancer drugs

## Abstract

*PTPN11* encodes the SHP2 protein tyrosine phosphatase that activates the mitogen-activated protein kinase (MAPK) pathway upstream of *KRAS* and MEK. *PTPN11*/Shp2 somatic mutations occur frequently in Juvenile myelomonocytic leukaemia (JMML); however, the role of mutated *PTPN11* in lung cancer tumourigenesis and its utility as a therapeutic target has not been fully addressed. We applied mass-spectrometry-based genotyping to DNA extracted from the tumour and matched the normal tissue of 356 NSCLC patients (98 adenocarcinomas (LUAD) and 258 squamous cell carcinomas (LUSC)). Further, *PTPN11* mutation cases were identified in additional cohorts, including TCGA, Broad, and MD Anderson datasets and the COSMIC database. *PTPN11* constructs harbouring *PTPN11* E76A, A72D and C459S mutations were stably expressed in IL-3 dependent BaF3 cells and NSCLC cell lines (NCI-H1703, NCI-H157, NCI-H1299). The MAPK and PI3K pathway activation was evaluated using Western blotting. *PTPN11*/Shp2 phosphatase activity was measured in whole-cell protein lysates using an Shp2 assay kit. The Shp2 inhibitor (SHPi) was assessed both in vitro and in vivo in a *PTPN11*-mutated cell line for improved responses to MAPK and PI3K targeting therapies. Somatic *PTPN11* hotspot mutations occurred in 4/98 (4.1%) adenocarcinomas and 7/258 (2.7%) squamous cells of 356 NSCLC patients. Additional 26 *PTPN11* hotspot mutations occurred in 23 and 3 adenocarcinomas and squamous cell carcinoma, respectively, across the additional cohorts. Mutant *PTPN11* significantly increased the IL-3 independent survival of Ba/F3 cells compared to wildtype *PTPN11* (*p* < 0.0001). Ba/F3, NCI-H1703, and NCI-H157 cells expressing mutant *PTPN11* exhibited increased *PTPN11*/Shp2 phosphatase activity and phospho-ERK1/2 levels compared to cells expressing wildtype *PTPN11*. The transduction of the *PTPN11* inactivating mutation C459S into NSCLC cell lines led to decreased phospho-ERK, as well as decreased phospho-AKT in the *PTPN11*-mutated NCI-H661 cell line. NCI-H661 cells (*PTPN11*-mutated, *KRAS*-wild type) were significantly more sensitive to growth inhibition by the PI3K inhibitor copanlisib (IC50: 13.9 ± 4.7 nM) compared to NCI-H1703 (*PTPN11*/*KRAS*-wild type) cells (IC50: >10,000 nM). The SHP2 inhibitor, in combination with the PI3K targeting therapy copanlisib, showed no significant difference in tumour development in vivo; however, this significantly prevented MAPK pathway induction in vitro (*p* < 0.0001). *PTPN11*/Shp2 demonstrated the in vitro features of a driver oncogene and could potentially sensitize NSCLC cells to PI3K inhibition and inhibit MAPK pathway activation following PI3K pathway targeting.

## 1. Introduction

Non-small cell lung cancer (NSCLC) accounts for approximately 80% of lung cancers, and the majority of these patients have metastatic disease at presentation, with <5% surviving for more than 5 years [1]. In recent years, the identification and rational therapeutic targeting of tumour genomic aberrations have revolutionized the care of patients with NSCLC [2]. However, in Western populations, only 20–25% of lung adenocarcinoma (LUAD) patients whose tumours harboured activating alterations in *EGFR*, *ALK*, or *ROS1* could be treated with targeted therapies [3]. Recently, an immune checkpoint blockade using monoclonal antibodies targeting the programmed death-1 (PD-1)/PD ligand 1 (PD-L1) pathway emerged as an effective therapeutic strategy in NSCLC [4]. Unfortunately, responses to immune checkpoint inhibitors are limited to only 18–23% of unselected NSCLC patients in the second-line setting [5,6]. Therefore, there is a clear need to unravel novel genetic drivers in NSCLC.

Protein tyrosine phosphatase non-receptor type 11 (*PTPN11*) encodes the Src homology 2 domain-containing protein tyrosine phosphatase 2 (Shp2): a ubiquitously expressed protein that plays multiple roles in cellular processes [7]. *PTPN11*/Shp2 is best known for its role in growth factors, cytokines, and other extracellular protein-induced signal transductions [8,9]. The crystal structure of *PTPN11*/Shp2 reveals that it is normally self-inhibited by the hydrogen bonding of the backside of the N-terminal SH2 (N-SH2) domain to the deep pocket of the PTP domain (see Appendix A). Ligands with phosphorylated tyrosine (pY) residues activate *PTPN11*/Shp2 by binding to SH2 domains (primarily N-SH2) and disrupting the inhibitory interaction between N-SH2 and PTP domains [10]. Germline mutations in *PTPN11* have been identified in the developmental disorder of Noonan’s syndrome [11]. The gain of function *PTPN11* in somatic mutations was found in 35% of juvenile myelomonocytic leukaemia (JMML) [12] and less frequently in acute leukaemia and adult solid tumours [13,14,15]. Most *PTPN11* mutations identified in childhood leukaemia are located in the N-SH2 domains. These mutations perturb the binding of the N-SH2 domain to the PTP domain and activate its phosphatase activity without stimuli [16].

Shp2 is required for the full activation of the Ras/MEK/MAPK pathway by most receptor tyrosine kinases (see Appendix A) [17,18]. The cells expressing dominant negative Shp2 exhibit defective Ras activation, indicating that Shp2 acts upstream of Ras [8]. A recent study by Ruess et al. demonstrated that *PTPN11* is redundant in the presence of constitutively active MEK1 or PI3K, indicating that Shp2 functions at the level of *KRAS* [19]. Although most studies implicate the role of Shp2 in the positive regulation of Ras/MEK/MAPK pathway signalling, Shp2 may also play a positive or negative role in the regulation of phosphatidylinositol-3-kinase (PI3K) signalling [20].

*PTPN11*/Shp2 is currently being investigated as a novel therapy target in some hematologic malignancies and solid tumours [12,21,22], and recently, Shp2 inhibition was been shown to reduce tumour burden significantly in vivo in a metastatic breast cancer model [23]. Moreover, Phase 1 and Phase 2 clinical trials are underway examining the Shp2 inhibitors TNO155 [24] (NCT04000529, NCT03114319, NCT04699188, NCT04185883), and RMC-4630 (NCT03634982, NCT05054725, NCT04916236) as monotherapies in solid tumours and through combinational approaches in *KRAS* and *EGFR*-mutated solid tumours. *PTPN11*/Shp2 can be targeted directly using inhibitors of *PTPN11*/Shp2 phosphatase and indirectly by inhibiting downstream proteins in the Ras/MEK/MAPK and PI3K pathways. Because most *PTPN11*/Shp2 inhibitors have limited potency against mutated Shp2 [25], along with non-specific effects on other PTPs, in this study, we wanted to investigate both direct and indirect targeting strategies using Shp2, MEK, and PI3K inhibitors in *PTPN11*-mutated NSCLC. In parallel, we wanted to determine the spectrum of somatic *PTPN11* mutations in NSCLC.

## 2. Results

### 2.1. PTPN11 Is Recurrently Mutated in NSCLC

We performed the sequenom genotyping of 258 surgically resected squamous cell lung carcinomas (LUSC) and 98 lung adenocarcinomas (LUAD). We identified 11 (3.1%) patients harbouring somatic *PTPN11* mutations: seven (2.7%) LUSCs and four (4.1%) LUADs (Figure 1). All 11 *PTPN11* mutations were validated using pyrosequencing and were confirmed to be somatic by demonstrating their absence in matched adjacent normal tissue using MassArray.

To broaden our cohort of *PTPN11*-mutated NSCLC, we reviewed independent lung cancer datasets. We identified seven *PTPN11*-mutated NSCLCs (all LUAD) in the MD Anderson Cancer Centre GEMINI database. In the TCGA and Broad datasets, we identified 13/408 (3.2%) and 3/178 (1.7%) *PTPN11*-mutated LUADs and LUSCs, respectively (Figure 1). A review of the COSMIC database revealed that three additional *PTPN11*-mutated LUADs and seven more *PTPN11*-mutated LUAD cases were included from the MD Anderson cohort. In total, this combined dataset included 37 *PTPN11*-mutated NSCLCs (27 LUADs and 10 LUSCs).

### 2.2. Spectrum of Somatic PTPN11 Mutations in NSCLC

In the combined dataset of 37 *PTPN11*-mutated NSCLC, E76A was the most common mutation detected, encompassing 19% (5/27) and 30% (3/10) of *PTPN11*-mutated LUAD and LUSC, respectively (Figure 1). G503V was identified in 7.5% (2/27) and 10% (1/10) of *PTPN11*-mutated LUAD and LUSC, respectively (Figure 1). A72D was found in 1/26 and 3/10 of *PTPN11*-mutated LUAD and LUSC, respectively (Figure 1). As discussed above, the crystal structure of *PTPN11*/Shp2 revealed an auto-inhibitory mechanism; the N-terminal SH2 domain of *PTPN11*/Shp2 directly blocked its enzymatic (PTP) active site. N-SH2 and PTP were shown to share a broad surface that was encoded by exons 3 and 13 of *PTPN11*. Hence, the mutations in exons 3 and 13 disturbed the binding of the N-SH2 domain to the PTP domain, leading to the loss of the auto-inhibitory mechanism. Therefore, mutations in exons 3 and 13 were thought to be activating mutations. Notably, in the combined set of *PTPN11*-mutated NSCLCs, 75% (26/37) of mutations occurred in exon 3 (16/37) or exon 13 (10/37), suggesting that most *PTPN11* mutations in NSCLC were functional. A detailed summary of *PTPN11* mutations in the 37 *PTPN11*-mutated NSCLCs is presented in Table 1.

Among the 37 *PTPN11*-mutated NSCLCs, ten had a concurrent *KRAS* mutation, five had a concurrent *PIK3CA* mutation, one had a concurrent *NRAS* mutation, and one patient had concurrent overlapping *PIK3CA* and *KRAS* mutations (Figure 1). Thus, almost half of the tumours had a co-existing somatic mutation in the PI3K and/or MAPK pathway. No alterations were observed in *EGFR*, *ALK*, *BRAF*, or *ERBB2* in these 37 tumours.

### 2.3. Clinical Characteristics of Patients with PTPN11-Mutated NSCLC

Robust clinical data were available on 34 out of 37 patients with *PTPN11*-mutated NSCLC. The clinical characteristics of these 34 patients are listed in Table 2. All patients were former or active smokers, with the majority (94%, 32/34) having more than a 10-pack-year smoking history. The median smoking history was 42 pack years (range, 1–113). The majority of patients were Caucasian (67%, 23/34) and male (67%, 23/34).

### 2.4. PTPN11 Mutations Promote Interleukin-3-Independent Cell Survival

Ba/F3 is a widely used pro-B cell line that is dependent on interleukin-3 (IL-3) for survival and proliferation but becomes IL-3 independent in the presence of an oncogene [26]. Leukaemia-associated *PTPN11* mutations were shown to promote the IL-3-independent survival of Ba/F3 cells [27]. In order to study the functional consequences of *PTPN11* mutations in NSCLC, we transduced Ba/F3 cells with the two most common somatic *PTPN11* mutations detected in our dataset: E76A and A72D, as well as with *PTPN11* wild type (WT) and an empty vector. We plated Ba/F3 cells in a medium without IL-3 in 3 × 96-well plates in triplicates and monitored them for survival. Compared to the wild-type and empty vector, *PTPN11* mutants promoted significant IL-3-independent Ba/F3 cell survival at 120 h (Figure 2): a phenotype suggestive of a transforming function associated with NSCLC-associated somatic *PTPN11* mutations.

### 2.5. PTPN11 Mutations in NSCLC Cause Gain of Function in Shp2 and Activate MAKP and PI3K Pathways Signaling

The biological functions of Shp2, the protein encoded by *PTPN11*, were mediated via its phosphatase activity. We hypothesized that *PTPN11* mutations in NSCLC would lead to the Shp2 gain of function, resulting in increased Shp2-phosphatase activity: a finding that was observed in leukaemia-associated *PTPN11* mutations [14]. Using MassArray analysis, we confirmed the mutational profile of the following NSCLC cells: NCI-H661 (*PTPN11*- exon 3 mutated (N58), *KRAS* and *PIK3CA* WT), NCI-H157 (*KRAS*-mutated, *PTPN11* and *PIK3CA* WT), NCI-H1703 (*PTPN11*, *KRAS* and *PIK3CA* WT) and Calu-3 (*PTPN11*, *KRAS*, and *PIK3CA* WT). We used an anti-Shp2 antibody conjugated to agarose immunoprecipitation (IP) beads to isolate SHP2 immunocomplexes from NSCLC cells and compared their Shp2 phosphatase activity. We found that *PTPN11*-mutated NCI-H661 cells had a significantly higher Shp2-phosphatase activity when compared to *PTPN11*-wild type NCI-H1703 and Calu-3 cells but not when compared to *KRAS*-mutated NCI-H157 cells, which had a similarly high Shp2-phosphatatse activity (Figure 3A). Using Sanger sequencing, we confirmed that NCI-H157 cells did not have a *PTPN11* mutation, which, if present, would have explained the high Shp2-phosphatase activity of these cells.

To further confirm the oncogenic activity of NSCLC somatic *PTPN11* mutations, we transduced E76A, A72D, WT *PTPN11*, and an empty vector in NCI-H1703 and NCI-H1299 NSCLC cells. As expected, we observed greater Shp2-phosphatase activity in NCI-H1299 and NCI-H1703 transduced with NSCLC *PTPN11* mutations compared to those transduced with wild type *PTPN11* (Figure 3B,C).

Next, we sought to test the effects of *PTPN11* mutations on cell signalling. In NCI-H157 and NCI-H1703 cells transduced with *PTPN11* mutations, we observed the significant activation of the MAPK and PI3K pathways, as evidenced by elevated phosphorylation of ERK1/2 (MAPK) and AKT, respectively (Figure 3D). Thus, *PTPN11* mutants in NSCLC activate multiple downstream pathways that are known to be important for transformation.

### 2.6. Somatic PTPN11 Mutations Sensitize NSCLC Cells to MEK and PI3K Inhibitors

We transduced NSCLC cell lines with the artificial *PTPN11*-inactivating mutation C459S: an Shp2 phosphatase domain mutation that resulted in catalytically inactive Shp2. Western blot analysis showed that NCI-H661 (*PTPN11* mutated), NCI-H157 (*PTPN11* WT), and NCI-H1703 (*PTPN11* WT) cells when transduced with *PTPN11* C459S had significantly reduced phospho-ERK (Figure 4). Importantly, the inactivation of *PTPN11*/Shp2 resulted in a decrease in phospho-AKT levels, specifically in NCI-H661 cells, which were *PTPN11* mutated, and had no effect on phospho-AKT levels in the *PTPN11*-WT NCI-H1703 and NCI-H157 cells (Figure 4). Thus, we predict that the PI3K pathway could play an important role in the oncogenic activity of mutated *PTPN11*.

### 2.7. SHPi Improves Response to MAPK and PI3K Targeting Therapies Regardless of PTPN11 Mutation Status

We hypothesized that the inhibition of Shp2 would sensitize *PTPN11*-mutated cells (NCI-H661) to the MAPK and PI3K targeting therapies refametinib and copanlisib. NCI-H661 and NCI-H1703 (*PTPN11*-WT) were treated with SHPi (10 µM), copanlisib (200 nM), or refametanib (10 µM) alone or in combination for 72 h, and their viability was assessed (Figure 5). A combination treatments significantly reduced the viability of cells, irrespective of their *PTPN11* mutation status, when compared to the vehicle control. H661 cells had a significant decrease in their cellular viability following a combination treatment with SHPi + refametinib (S + R) of 42.26 ± 13.13% (*p* = 0.0341) compared to the vehicle control (VC), and although it was not significant, showed a trend of increased sensitivity compared to refametinib alone (31.86 ± 7.74%). This same trend was also seen in copanlisib-treated cells, where a decrease in viability (6.3 ± 3.55%) was improved in the combination setting (31.63 ± 10.28%). However, *PTPN11*-WT cells showed an overall improved sensitivity (though not significantly) compared to *PTPN11*-mut cells. NCI-H1703′s cellular viability was decreased by 42.36 ± 3.51% (*p* = 0.0277) and 37.81 ± 14.16% (*p* = 0.0561) following copanlisib and refametanib treatment (respectively). The cellular viability of NCI-H1703 was further decreased (significantly when compared to VC) in the combination settings, where decreases of 56.9 ± 4.39% (*p* = 0.0035) and 58.17 ± 4.56 (*p* = 0.0029) was shown in SHPi + copanlisib (S + C) and SHPi + refametanib (S + R), respectively. Therefore, can predict that Shp2 inhibition may sensitize NSCLC cells to therapies of related pathway targets.

### 2.8. Shp2 Inhibition Prevents Positive Feedback Activation following PI3K Targeted Therapy in PTPN11-Mutated Setting

Having shown that cellular viability appeared to be sensitized to PI3K and MAPK targeting in combination with Shp2 inhibition, we next sought to investigate whether dual treatment impacted the downstream pathway members via Western blotting. Both H661 and H1703 cells were treated with SHPi (10 µM), copanlisib (200 nM), and refametinib (10 µM) for 72 h, following which the proteins were extracted, quantified, and imaged via Western blotting for phosphorylated AKT (pAKT) and phosphorylated ERK1/2 (pERK) before being densitometrically analysed (relative to Actin and Total AKT/ERK). Although there were significant decreases in *PTPN11*-WT NCI-H1703 cell pAKT following copanlisib treatment and pERK following refametinib treatment, these differences were not significantly altered by the combination settings (Figure 6). However, in *PTPN11*-mutated NCI-H661 cells, there was a demonstrated increase in pERK following copanlisib treatment, suggesting the activation of the MAPK pathway, but this was significantly attenuated in the presence of SHPi when in combination with copanlisib (*p* ≤ 0.0001) (Figure 6). Thus, we can suggest that SHPi treatment in combination with PI3K pathway targeting agents might prevent the positive feedback activation of downstream MAPK pathway members.

### 2.9. PTPN11-Mutated NSCLC Cells Demonstrate Aggressive Phenotype In Vivo

Given our in vitro findings to date, we hypothesized that *PTPN11*-mutated H661 cells could respond to a combination therapy regime of the SHPi and PI3K targeting therapy copanlisib in vivo. We utilised the Chick chorioallantoic membrane in vivo assay as a non-rodent xenograft generating model. In total, 2 × 10^6^ H661 cells were implanted and xenografts were generated and treated before being extracted for analysis. Overall, our model demonstrated no significant impact on the xenograft macroscopic visibility upon extraction for any treatment group (Figure 7). However, we observed that the model was severely impacted by the naturally aggressive nature of H661 cells in vivo. This was demonstrated through an unusually high loss of vehicle control and untreated eggs, despite rigorous procedural controls to prevent this. Xenografts treated with any inhibiting agent appeared to be more likely to survive than those that did not (observation not analysed). We could, therefore, predict that given this observed aggressive nature, in vivo modelling with this *PTPN11*-mutated cell line could prove difficult for interrogation in other settings.

## 3. Discussion

The identification of somatic aberrations in *EGFR*, *ALK,* and *ROS1* has led to a paradigm shift in the treatment of NSCLC, with the development of specific molecular treatments to target these mutations. These targeted therapies have significantly improved in their therapeutic efficacy, often in conjunction with decreased toxicity. A study by the Lung Cancer Mutation Consortium demonstrated a median survival of 3.5 years for patients with an oncogenic driver mutation who received genotype-directed therapy compared with 2.4 years for patients with an oncogenic driver mutation who did not receive genotype-directed therapy [28]. However, only a minority of cancer patients have tumours that possess targetable molecular aberrations currently. Therefore, there is an urgent need to augment the successes to date with additional therapies that can target alternate molecular aberrations and signalling pathways.

Germline missense mutations in *PTPN11* are responsible for 50% of cases of Noonan syndrome: an autosomal dominant trait characterized by developmental disorders [11]. Somatic mutations in *PTPN11* are found in 35% of sporadic juvenile myelomonocytic leukaemia (JMML) [12] and at a lower frequency in childhood myelodyplastic syndrome, B-ALL, and acute myologenous leukaemia (AML), as well as adult AML [13,14,15]. Somatic *PTPN11* mutations have also been reported, albeit more rarely, in solid tumours, including colorectal, melanoma, and lung [13,14,15]. Although implicated, the role of *PTPN11* mutations in lung cancer tumorigenesis and its utility as a therapeutic target has not been fully addressed in lung cancer. Here, we have shown that somatic mutations in *PTPN11* occur in the tumours of 3.1% of NSCLC patients. The frequency of *PTPN11* mutations was higher in LUAD compared to LUSC (4.1% vs. 2.7%). This was confirmed in additional lung cancer datasets. Interestingly *PTPN11* mutations were present across all tumour stages. Given the varied tumour staging within this cohort, the further elucidation of *PTPN11* mutations present in individual NSCLC stages across a larger population may prove important for interrogating the best targetable clinical benefit.

The most common *PTPN11* mutations identified in the combined datasets were E76A, A72D, and G503V, which occurred in exons 3 and 13 and were likely to destabilize the inactive conformation of *PTPN11*/Shp2 without altering Shp2′s catalytic capability. The functional analysis confirmed that E76A and A72D were associated with relatively high Shp2 phosphatase activity and enhanced the IL-3 independent survival of Ba/F3 cells, indicating that targeting these mutations could be a potential therapeutic target.

We have shown that somatic *PTPN11* mutations increase proliferation, activate the Shp2 phosphatase domain, and promote the activation of the PI3K and MAPK pathways in NSCLC cell lines, which is consistent with prior studies in leukaemia [8,9,11,14,29,30]. Efforts are underway to develop *PTPN11* inhibitors as novel targeted therapies; however, these inhibitors often have poor bioavailability and low sensitivity [31], although newer *PTPN11* inhibitors, including SHP099, which stabilize *PTPN11* in an auto-inhibited conformation have been developed to circumvent this [32]. However, SHP099 was found to be ineffective in *KRAS*-mutant or *BRAF*-mutant cancer cell lines [32], suggesting that the signalling and phenotypic effects of *PTPN11* mutations and optimal targeting strategies may be impacted by co-existent mutations, particularly *PIK3CA* and *RAS* mutations. Indeed, we have found that almost half of the *PTPN11*-mutated NSCLCs had a co-existing somatic mutation in the PI3K and/or MAPK pathway. Our results showed that PI3K and MAPK pathways were activated in *PTPN11* mutated cell lines, and the transduction of the *PTPN11* mutated H661 cell line with an inactivating *PTPN11* mutation that significantly reduced PI3K and MAPK pathway activation. Furthermore, the combination of SHPi (SHP099) with PI3K targeting therapy prevented MAPK activation, confirming that the PI3K and MAPK pathways often play an important role in the oncogenic activity of *PTPN11* mutated NSCLC. Additionally, given the frequent presence of co-existing *PIK3CA* and *RAS* mutations in *PTPN11*-mutated human NSCLC samples, we hypothesized that targeting the PI3K or MAPK pathway could be a rational therapeutic approach in *PTPN11*-mutated NSCLC. We used the PI3K inhibitor copanlisib (BAY-806946) and the MEK inhibitor refametinib (BAY86-9766), as these drugs have already been administered to patients with solid tumours in phase I/II clinical trials [33,34,35,36].

We observed the sensitivity of *PTPN11*-mutated NCI-H661 NSCLC cells to growth inhibition by the PI3K inhibitor copanlisib and the MEK inhibitor refametinib. Notably, we confirmed that NCI-H661 cells did not have a PI3K pathway mutation or PTEN loss, both of which, if present, could explain this cell line’s sensitivity to PI3K inhibition. Importantly H661 cells were approximately as sensitive to copanlisib as the *PIK3CA* mutated BT474 breast cancer cell line, which is the most sensitive cell line to copanlisib that we have found to date [37]. Both *PTPN11* mutations and *KRAS* mutations resulted in the hyperactivation of the MAPK pathway, which could be blocked with a MEK inhibitor. However, NCI-H661 cells were less sensitive to the MEK inhibitor refametinib, which is likely because MEK inhibition can be associated with Shp2 activation [38]. Combining targeted therapies such as MEK inhibitors and Shp2 inhibitors may be a treatment possibility; however, increased toxicities are likely to be a concern. Furthermore, given that *KRAS* is known to play a major role in altering the metabolic landscape of NSCLC, it is important to elucidate whether Shp2 can also cause metabolic alterations in this setting and whether these alterations are impacted by the presence of Shp2 inhibition [39,40].

Finally, crosstalk between the PI3K and MAPK pathway has been well established, and the inhibition of one oncogenic pathway activating the other, potentially through negative feedback loops, has frequently been demonstrated [41,42,43]. We observed a similar phenomenon of alternative activation following PI3K and MAPK inhibition, which was attenuated through the addition of an Shp2 inhibitor. Given the mechanistic role that Shp2 plays in both pathways, it is possible that Shp2 inhibition, in conjunction with either therapy approach, may act synergistically to increase its sensitization to therapy through the prevention of alternative oncogenic pathway activation. Furthermore, this synergistic effect may indeed be demonstrated irrespective of *PTPN11′s* mutation status and warrants further investigation in PI3K or RAS-driven NSCLC settings, where either PI3K or MAPK targeting is appropriate. 

Overall, our data suggest that *PTPN11* mutations may sensitize NSCLC cells to PI3K inhibition in particular, which could potentially be tested as a new personalized treatment strategy and could improve the outcomes of as many as 3% of NSCLC patients with *PTPN11*-mutated tumours who currently do not have molecularly aberration-directed targeted therapy options available for their treatment. Furthermore, our findings also suggest that a combination therapy approach using Shp2 inhibition and PI3K dual targeting may work synergistically to prevent the downstream activation of alternative oncogenic pathways. However, carefully designed clinical studies are necessary to determine the toxicity profile of these combinations. While the percentage of NSCLC patients with somatic *PTPN11* mutations in their tumours is relatively low, it is roughly equivalent to the frequency of *ALK* translocations, the targeting of which has revolutionized the treatment of NSCLC. 

## 4. Materials and Methods

### 4.1. Patient Data

The clinical cases of non-small cell lung cancer (NSCLC) treated at two institutions (St James’s University Hospital and Beaumont Hospital, Dublin) between January 2000 and June 2004 were identified for this review. Clinical characteristics, including age, gender, race, smoking history, clinical stage, and treatment, were gathered from patients’ records. All patients involved in the study provided written informed consent for tissue banking and for genomic studies in accordance with the institutional Research Ethics Committee, and their data were anonymized according to regulatory requirements.

### 4.2. Tissue Procurement and Mutational Profiling

Tumour specimens were obtained as the standard of care for clinical management. All specimens included in this study were from surgically resected NSCLC tumours. Genomic DNA was extracted from tumour samples and matched with adjacent normal tissue using standard procedures. The tumours were genotyped by the Agena MassArray system (Agena Biosciences, San Diego, CA, USA). Briefly, the samples were tested in duplicate using a series of multiplexed assays that were designed to interrogate a total of 547 single nucleotide mutations in 48 oncogenes (listed in Appendix A). Genomic DNA amplification and single base pair extension steps was performed using specific primers designed with Agena Assay Designer v3.1 software. The allele-specific single base extension products were then quantitatively analysed using matrix-assisted laser desorption/ionization-time of flight mass spectrometry (MALDI-TOF MS) on the Agena MassArray Spectrometer. All automated mutation calls were confirmed by a manual (visual) review of the spectra. Somatic *PTPN11* mutations detected through MassArray analysis were validated using pyrosequencing. Additionally, sizing electrophoresis and fluorescence in situ hybridization (FISH) were performed to detect EGFR deletions and ALK gene rearrangements.

### 4.3. Identification of PTPN11 Mutations in Additional NSCLC Patients’ Datasets

To evaluate *PTPN11* mutations in a broader population of NSCLC patients, the MD Anderson Cancer Centre Lung Cancer Moon Shot GEMINI database, which includes 2515 NSCLC patients (IRB approval PA16-0061), was reviewed. As part of routine clinical care, the tumours from patients who enrolled in the GEMINI database were subjected to targeted next-generation sequencing with a minimum of 46 cancer-related genes. In addition, >10,000 Foundation Medicine NSCLC clinical cases were interrogated for *PTPN11* mutations. The detailed laboratory and computational methods used in the Foundation Medicine assay (FoundationOne^®,^ Cambridge, MA, USA) have been reported [44]. Next-generation sequencing data, made publicly available through the cBioportal (TCGA and Broad) [45,46] as well as the COSMIC database, were also reviewed [47].

### 4.4. Cell Lines

Cell lines were purchased from the American Type Culture Collection or were a kind gift from Ulla Knaus (Conway Institute, University College Dublin). The NCI-H1703, NCI-H157, NCI-H1299, and NCI-H661 NSCLC cells were cultured in RPMI 1640 (Sigma) supplemented with 10% fetal bovine serum. Calu-3 NSCLC cells were cultured in Eagle’s Minimum Essential Medium (Sigma) and supplemented with 10% fetal bovine serum. Ba/F3 cells, an IL-3-dependent murine pro-B cell line, were cultured in RPMI 1640 supplemented with a 10% fetal bovine serum and 1.0 ng/mL recombinant mouse IL-3. All cell lines used in this study were fingerprinted to authenticate the cell lines (Source BioScience, Nottingham, UK).

### 4.5. Generation of Stable Cell Lines

A full-length *PTPN11* cDNA was inserted at the EcoRI site of the pBabe vector (Bob Weinberg, Addgene plasmid, #1764). The single nucleotide change resulting in the A72D substitution was introduced by site-directed mutagenesis (QuikChange II XL Site-Directed Mutagenesis Kit, Agilent Technologies). The constructs expressing *PTPN11* wild type (WT), E76A, and C459S (Ben Neel, Addgene plasmids 8329, 8331, and 8382, respectively) were obtained. Retroviruses were produced by the transfection of HEK-293T cells with plasmids encoding *PTPN11* WT, E76A, A72D, C459S, and an empty vector (EV) using the FuGENE (Promega) reagent. Culture supernatants containing a retrovirus were collected 48 h post-transfection and used to infect target cells, which were selected in puromycin or neomycin. All constructs were validated by Sanger sequencing.

### 4.6. Cell Proliferation and Growth Assays 

Cell lines were mycoplasma tested before and after the in vitro experiments. The Shp2 inhibitor (SHPi) (SHP099), PI3K inhibitor copanlisib (cop) and the MEK1/2 inhibitor refametinib (ref) were obtained from Selleckchem and stocks (10 mM SHPi, 10 mM refametinib, 5 mM copanlisib;) were prepared in 100% DMSO, 100% DMSO and 100% DMSO with 10 mM TFA, respectively. The growth inhibition of Ba/F3 and NSCLC cells was assessed by a CellTiter 96 AQueous One Solution Cell Proliferation Assay (Promega, Madison, WI, USA) and acid phosphatase assays, respectively. The data were graphically displayed using GraphPad Prism version 6.00 for Windows (GraphPad Software). The curves were fitted using a nonlinear regression model with a sigmoidal dose-response. The cellular viability of *PTPN11* WT and mutated cells was assessed by the Alamar Blue assay, and fluorescence was measured at 610 nm using VICTOR^TM^ X3 2030 Multilabel Plate Reader. Statistical analysis was performed using Graphpad Prism 8.4.3. Statistical significance was calculated following the software-recommended tests and post-tests and is described in individual figure legends.

### 4.7. Western Blot Analysis

Cell lysates were prepared using a cell extraction buffer supplemented with protease and phosphatase inhibitors (Sigma-Aldrich, St. Louis, MO, USA). Proteins were electrophoresed by SDS–PAGE and transferred to a nitrocellulose membrane using a semi-dry transfer apparatus (Bio-Rad Laboratories, Hercules, CA, USA). After incubation with 5% nonfat milk in TBST (10 mmol/L Tris, pH 8.0, 150 mmol/L NaCl, 0.5% Tween 20) for 60 min, the membranes were incubated with primary antibodies (See Appendix A for a full list of antibodies) at 4 °C overnight. The membranes were washed three times and incubated with horseradish peroxidase (HRP)-conjugated secondary antibodies. Blots were developed after the ECL-based chemiluminescence reaction through film exposure. The average mean data (±standard error of the mean (SEM)) values were calculated and graphed using Graphpad Prism 8.4.3, Experimental data (*n* = 3) were statistically tested as described in the figure legends (* *p* < 0.05, ** *p* < 0.01, *** *p* < 0.001, **** *p* < 0.0001)

### 4.8. Shp2 Phosphatase Activity Assay

Shp2 phosphatase activity was measured in whole-cell protein lysates using a Shp2 assay kit (R&D Systems, Minneapolis, MN, USA) according to the manufacturer’s instructions. Briefly, 100 µg of the total protein was immunoprecipitated with *PTPN11*/Shp2-specific beads, incubated with a synthetic phosphopeptide as the substrate, and phosphate release was measured. The absorbance of the reaction mixture was measured at 620 nM.

### 4.9. Chick Chorio-Allantoic Membrane (CAM) Assay

Fertilised chick eggs were obtained from Shannon Vale Hatcheries (Co. Cork, Ireland) and were wiped gently with 70% EtOH before being placed in a 37 °C incubator with 0% CO_2_ (day 0). On day 3, a small puncture hole was made in the bottom right quadrant of the shell, and ~3 mL of albumin was removed using an 18G syringe. The hole was recovered with a semi-permeable membrane. A window was then cut into the top of the shell and recovered with a semi-permeable membrane. On day 7, the semi-permeable membrane windows were reopened, and a silicon ring (taken from sterile cryovials and washed in EthOH, followed by ddH_2_O) was placed directly onto the CAM surface. A total of 2 × 10^6^ untreated H661 cells were pelleted and resuspended in 25 µL of RPMI-1640 plus 25 µL of matrigel. The 50 µL cell: matrigel suspension was slowly added within the bounds of the silicon ring. The window was re-covered with a fresh semi-permeable membrane, and the eggs were replaced at 37 °C. The eggs were split into four groups for treatment with either vehicle control, SHPi, Copanlisib, or SHPi + Copanlisib (S + C). On days 10–14, developing tumours for the corresponding treatments were treated with SHPi (10 µM) or vehicle control. On day 10, the corresponding eggs were treated with Copanlisib (200 nM). The treatments were added by adding 15 µL directly onto the xenografts sitting within the silicon ring. On day 14, the embryos were sacrificed, and the visibility of tumours was noted for each egg. The xenografts were extracted, washed in ddH_2_O, fixed in 10% formalin, paraffin-embedded, sectioned, and sent for hematoxylin and eosin staining.

### 4.10. Statistical Analysis

All analyses were carried out using GraphPad Prism 8.4.3, where data were graphed as the mean ± standard error of the mean (SEM). Data requiring statistical analysis are described in more detail within individual figure legends.

## Figures and Tables

**Figure 1 ijms-24-10545-f001:** *PTPN11* mutation occurrence and co-occurrence alongside other mutations in both lung adenocarcinomas and squamous cell carcinomas. *PTPN11* mutation occurrence rate across the genotyped tumour tissue of NSCLC patients (*n* = 356) and TCGA data (*n* = 586) (**A**). Oncoprint shows the gene alterations in each individual with *PTPN11*-mutated NSCLC (*n* = 37), focusing on known cancer-related genes. Each box represents a patient; genes and corresponding alteration frequencies are listed (**B**). The type of *PTPN11* mutation occurring across both adenocarcinomas (LUAD) and squamous cell carcinoma cohorts (LUSC) is displayed (*n* = 37) (**C**).

**Figure 2 ijms-24-10545-f002:** *PTPN11* mutations promote IL-3-independent survival of Ba/F3 cells. The stable expression of E76A and A72D *PTPN11* mutations promoted IL-3 independent survival of Ba/F3 cells. Ba/F3 cells transduced with indicated vectors (EV = empty vector, pBabe) were plated in the absence of IL-3. Viable cells were determined at 0 h, 24 h, 48 h, 96 h, and 120 h. Results are representative of three independent experiments. Data are represented as mean ± s.e.m. **** *p* < 0.0001.

**Figure 3 ijms-24-10545-f003:** *PTPN11* mutations result in elevated SHP2-phosphatse activity and activate MAPK and PI3K pathway signalling. Serum-starved cells treated with 100 ng/mL EGF for 5 min were lysed, and SHP2 was immunoprecipitated from whole cell lysates. Immunoprecipitate was used to determine phosphatase activity, as described in the materials and methods. (**A**) SHP2-phosphatse activity in H661 (*PTPN11*-mutated) compared to H1703 (*PTPN11*-WT), Calu-3 (*PTPN11*-WT), and H157 (*PTPN11*-WT, *KRAS*-mutated). Shp2-phosphatase activity in H1701 (**B**) and H1299 (**C**) cells transduced with indicated *PTPN11* mutations. Data are represented as mean ± s.e.m. (*n* = 3). (**D**) NCI-H1703 and NCI-H157 cells were transduced with wildtype or mutated *PTPN11*: a serum starved and stimulated with an epidermal growth factor (100 ng/mL) for 5 min. p-ERK1/2—phospho-ERK 1/2; t-ERK1/2—total ERK 1/2; p-AKT ser 473—phospho-AKT (phosphorylated at serine 473); t-AKT—total AKT. Blots are representative of 3 independent experiments. (NS = No Significance, * *p* < 0.05, ** *p* < 0.01, *** *p* <0.001)

**Figure 4 ijms-24-10545-f004:** *PTPN11*/Shp2 inactivation with the *PTPN11*/Shp2 phosphatase mutation C459S. NCI-H157, NCI-H1703, and NCI-H661 cells were transduced with *PTPN11* C459S: serum starved and stimulated with an epidermal growth factor (100 ng/mL) for 5 min. Parental cells were transfected with an empty vector. Phosphorylated-ERK 1/2 (pERK); total ERK 1/2 (tERK); Phosphorylated AKT (phosphorylated at serine 473) (pAKT); total AKT (tAKT). Blots are representative of 3 independent experiments.

**Figure 5 ijms-24-10545-f005:** SHP2 inhibitor (SHPi) improves the response to MAPK and PI3K pathway targeting therapies. H661 and H1703 cells plated and after 24 h were treated with SHPi (10 µM) daily or Cop (20 nM) or Ref (10 µM) alone or in combination with SHPi. 72 h following treatment, an Alamar Blue cell viability assay was performed (A). Data (*n* = 3 independent experiments) are expressed as mean ± SEM. Statistical significance was determined using one-way ANOVA correcting for multiple comparisons using Tukey’s test and reporting adjusted *p* values. (* *p* < 0.05, ** *p* < 0.01).

**Figure 6 ijms-24-10545-f006:** SHPi prevents feedback activation of MAPK and PI3K pathways following targeted treatments. H661 and H1703 cells seeded at a density of 2 × 10^5^ cells per well and were treated 24 h later with SHPi (10 µM) daily or Cop (20 nM) or Ref (10 µM) alone or in combination with SHPi for 72 h. The expression of pAKT and pERK was analysed using Western blotting (representative images displayed) (**A**,**B**), and densitometry was quantified using either ImageJ or ImageLab software 5.2.1. (The volume intensity of the band relative to actin was matched to the total controls and displayed relative to the vehicle control). H661 Results are expressed as mean ± SEM (*n* = 4) (**C**,**D**), and H1703 results are expressed as mean ± SEM (*n* = 3) (**E**,**F**). Statistical significance was determined using one-way ANOVA correcting for multiple comparisons using Tukey’s test and reporting adjusted *p* values. (* *p* < 0.05, **** *p*< 0.0001).

**Figure 7 ijms-24-10545-f007:** *PTPN11* and PI3K targeting treatments do not alter tumour formation or invasion in a chick embryo xenograft model. 2 × 10^6^ H661 cells were implanted into the CAM according to the assay schedule (**A**) on day 7 of embryonic development. Following 72 h of tumour establishment, developing tumours were treated in situ. SHPi (10 µM) treatments were added daily, and Cop (20 nM) treatments were added once on day 10. On day 14, tumour visibility was noted as either visible (VT) or non-visible (NVT). Statistical significance was determined using Fisher’s exact test, and no statistical significance was found (**B**) On day 14, xenografts were excised with the silicon ring, formalin-fixed, and stained for H&E (**C**,**D**). Areas of tumour were denoted by black arrows, and CAM areas were demonstrated by red arrows and matrigel was denoted by grey arrows. Images were collected an EVOS m5000 microscope with EVOS imaging software at 4× and 20× magnification (**D**).

**Table 1 ijms-24-10545-t001:** *PTPN11*-mutated NSCLC cases, (total *n* = 37, LUAD = 27, LUSC = 10), identified in combined datasets (this study, GEMINI, TCGA, Broad Institute, and COSMIC database).

Amino Acid Mutation	Nucleotide Substitution	Frequency N(%)	Exon
p.N58S	c.173A > G	1 (2.5%)	3
p.G60D	c.179G > A	1 (2.5%)	3
p.A72D	c.215C > A	4 (11%)	3
p.E76K	c.226G > A	1 (2.5%)	3
p.E76A	c.227A > C	8 (21.5%)	3
p.M82V	c.244A > G	1 (2.5%)	3
P.E225 *	c.673G > T	1 (2.5%)	7
p.E313 *	c.937G > T	1 (2.5%)	8
p.E412G	c.1235A > G	1 (2.5%)	11
p.Y418 *	c.1250G > T	1 (2.5%)	11
p.D425Y	c.1273G > T	1 (2.5%)	11
p.H426R	c.1277A > G	1 (2.5%)	11
p.V428M	c.1282G > A	1 (2.5%)	11
p.A452S	c.1354G > T	1 (2.5%)	12
p.G483D	c.1448G > A	1 (2.5%)	12
p.Q500E	c.1498C > G	1 (2.5)	13
p.S502L	c.1502A > G	3 (8%)	13
p.G503V	c.1508G > T	3 (8%)	13
p.G503R	c.1508G > A	1 (2.5%)	13
p.M504I	c.1512G > A	1 (2.5%)	13
p.Y521F	c.1562A > T	1 (2.5%)	13
p.D551N	c.1651G > A	1 (2.5%)	14
p.D556N	c.1666G > A	1 (2.5%)	14

**Table 2 ijms-24-10545-t002:** Clinical characteristics of patients with *PTPN11* mutations (*n* = 34).

Clinical Characteristics	N (%)
**Median age**	65 (range, 53–86)
**Gender**	
**Male**	23/34 (67%)
**Female**	11/34 (33%)
**Smoking History**	
**<10 pack years**	2 (6%)
**>10 pack years**	32 (94%)
**Pack years**	Median 42 (range, 1–113)
**Race**	
**Caucasian**	23 (67%)
**African American**	1 (3%)
**Unknown**	10 (30%)
**Stage at diagnosis**	
**IA**	2 (6%)
**IB**	10 (30%)
**IIA**	5 (14%)
**IIB**	6 (18%)
**IIIA**	5 (14%)
**IIIB**	4 (12%)
**IV**	2 (6%)

## Data Availability

The original contributions presented in this study are included in the artle/Appendix A. Further inquiries can be directed to the corresponding author.

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
