# Peer review of "Protein Tyrosine Phosphatase Non-Receptor 11 (PTPN11/Shp2) as a Driver Oncogene and a Novel Therapeutic Target in Non-Small Cell Lung Cancer (NSCLC)"

_ijms, 2023, doi:10.3390/ijms241310545_

Round 1

Reviewer 1 Report

An interesting article by Dr. Richards and the group elaborates on the role of SHP2 as a Driver Oncogene in NSCLC and explores the therapeutic intervention by utilizing small molecule inhibitors. Since the discovery of KRAS G12C and SHP2 inhibitors, the field has been moving fast and has a huge translational aspect. This manuscript holds novelty and will significantly impact the NSCLC field. A few things must be addressed before it is ready for acceptance; they are as follows:

1. It's known that KRAS is one of the significant oncogenes involved in NSCLC, and also SHP2 is involved in regulating KRAS-mediated pathways. It has been discussed earlier that oncogenic KRAS plays a major role in altering the metabolic landscape of  NSCLC (PMID: 33870211 and PMID: 28570035). Authors must add a few lines discussing this as one of the future aspects of elucidating SHP2's role in metabolic alterations in NSCLC.

2. Authors must mention about the current SHP2 inhibitors- for example TNO155, SHP2 inhibitor from Bridbio, Revolution medicines. This will add up the significance and value of SHP2 in NSCLC rather as an additive to G12C inhibitors. Based on the findings of this manuscript, those inhibitors can be utilized separately as a single arm in NSCLC.

3. Authors must add a model depicting the role of SHP2 in the signaling aspect specifically in NSCLC. This will again signify the impact of SHP2 as a driver oncogene in NSCLC. 

Reviewer 2 Report

In this interesting paper the authors describe PTPN11/Shp2 as a driver oncogene that can be potentially targeted. While it has been described already in other tumor entities (such as melanoma) and hematologic diseases, this is the first analysis in NSCLC. The paper is methodologically clear and well written so that – from my part – there are only some minor comments.

Patients (including Table 2)

This small cohort seems to be very heterogeneous. There is a huge difference – in terms of tumorbiology – between stage Ia and IV patients. Thus if you analyze material from patients who differ clinically/biologically with the idea of finding a drugable target the result is blurred by the diversity of the tumor tissue. By focusing on one tumor entity, e.g. stage III NSCLC, one would enhance the tanslational/clinical validity of the results. Please comment.

Discussion

Lines 366-368: What sort of therapy (surgery, radiotherapy, chemotherapy, immunotherapy) did the patients receive? How do the authors envisage a combination of PTPN11/Shp2 inhibitors with standard therapy regimens especially in regard to toxicity?

Line 376: affect should be effect

No major language concerns.

Round 2

Reviewer 1 Report

All concerns addressed and ready for acceptance. 

Reviewer 2 Report

I have nothing to add.